# Towards Personalized Parameter Generation via Data-Conditioned Mapping

## Abstract

Fine-tuning is the dominant strategy for adapting pre-trained models. However, it requires bulky gradient computation and model updates, which prevent real-time personalization. Even efficient variants such as LoRA incur a non-negligible latency and computation overhead. We explore a radically different approach: instead of training, we generate model parameters conditioned on data. Inspired by parameter generation via diffusion, we introduce a data-conditioned parameter generator that instantly produces personalized weights. In few-shot settings, our method outperforms LoRA in both performance and efficiency, making adaptation in less than one second. This demonstrates a feasible path toward practical, real-world, real-time model personalization.

## 1 Introduction

In the past decade, pretraining large models with massive data has become a prevalent paradigm (Devlin et al., 2019; Raffel et al., 2020; Achiam et al., 2023; Dosovitskiy et al., 2021a; He et al., 2021). Fine-tuning is the dominant strategy for adapting pre-training models. Adapted pre-trained models keep refreshing state-of-the-art results in various subdomains(Dosovitskiy et al., 2021b; He et al., 2021). However, this paradigm involves gradient computation and model updates, which incur a non-negligible latency and overhead for real-time model personalization.

Instead of relying on training or fine-tuning, a growing line of research explores neural network parameter generation through inference-only computation during deployment. Recent advances in this direction have substantially improved the *quality* and *scalability* of synthesized parameters. In particular, Schürholt et al. (2024); Wang et al. (2025) demonstrate the ability to generate models with hundreds of millions of parameters. These developments mark a promising step toward the vision of "AI creating AI" and open up new possibilities for real-time model personalization.

Parameter generation can be viewed as a specialized form of generative AI (GenAI) that learns to model the distribution of neural network parameters. While widely known GenAI applications such as *ChatGPT* (Achiam et al., 2023), *DALL·E* (Ramesh et al., 2021), and *Sora* (OpenAI, 2024) readily produce personalized content for end users, existing parameter generation approaches (Ha et al., 2016; Schürholt et al., 2022b; Wang et al., 2024; 2025) remain limited in their ability to synthesize personalized parameters (Figure 1). In this work, we carefully study this challenge and propose a practical solution.

The success of modern generative AI models is largely driven by two critical factors: (i) diverse training data and (ii) effective mapping from inputs to generated contents (*i.e.*, generator). Inspired by their success, we craft a novel and diverse training dataset for data-conditioned parameter generation, and design a model structure specialized for the task. Different to prior works(Wang et al., 2024; Soro et al., 2024b; Schürholt et al., 2022b), which only study parameters alone with infrequently used conditions, the new training dataset spreads across diverse real data and the corresponding model parameters. This requires a model structure adapted to the specialized generation requirements.

To fulfill the requirements, we make the following designs.

- *Structure-aware arrangement.* Previous parameter generation methods, such as RPG (Wang et al., 2025), tokenize different layers separately with a fixed input token shape into a sequence. However, this approach disrupts the original distribution of weights within the

(a) Popular GenAI tools can easily synthesize customized contents aligned with specific human prompts.

(b) Existing parameter generation works usually face challenges in synthesizing personalized parameters.

Figure 1: Comparisons between text, image, video, and parameter generation. Generating personalized neural network parameters is still a challenging research problem.

same layer, where the row-column geometric relationships are interrupted. This leads to a difficulty in representing layer-wise parameter distributions. To verify it, we visualize the cumulative normalized singular values curve in Figure 2. As the weight matrix is broken, it requires more singular values to capture the information, increasing the modeling difficulty. Instead, we design a structure-aware arrangement of parameters as the learning objective, keeping these relationships within the same layer and organizing parameters according to their types. This allows for easier learning with our method.

- *Semantic-parameter decoder.* The decoder's architecture is straightforward. As we are learning a mapping from data to parameters, we utilize a pretrained model to extract features from the given data (*e.g.*, texts or images), followed by a deformation module to generate data prototypes. These prototypes are then processed through cascaded decoder blocks that employ hybrid directional convolutions to extract rich data representations. The decoder output is fed into a projection layer that predicts the final parameters. We optimize this entire pipeline using a weighted MSE loss, with weights determined by parameter sensitivities.

In general, our approach establishes a novel paradigm for obtaining training models by mapping data directly into corresponding parameters. It broadens the parameter generation practices. The efficiency of our method is remarkable – once the decoder is trained, we can synthesize diverse large-scale parameters within seconds. Notably, these synthesized parameters yield promising few-shot results themselves. They could also serve as excellent initialization points for minimal-cost fine-tuning to further enhance performance. Empirical evidence confirms the advances of this initialization across extensive experiments. Specifically, in object detection, initialization using our method reduces training costs to merely 10% (compared to conventional training) by simply fine-tuning the generated models for a few steps.

## 2 METHODOLOGY

### 2.1 OVERVIEW

In this section, we present the details of our training data collection and proposed framework for personalized parameter generation. To preserve each layer's parameter matrices in their natural shapes, we apply *structure-aware arrangement* to normalize the parameters and divide them into normalized weights, normalized biases, and corresponding statistical measures. This arrangement reduces the learning difficulty of parameter distributions. Additionally, we employ a *semantic-parameter decoder* that transforms input data into generated parameters through data representation deformation, cascaded decoding blocks, and multi-task parameter projection. Finally, we adopt learnable loss weights across different layers based on their sensitivities to input variations. These weights are used to weight the MSE loss between predicted parameters and structure-aware arranged ones.

### 2.2 TRAINING DATA COLLECTION

To ensure rich diversity across multiple tasks, we collect hundreds to thousands of specialized models and construct data-parameter pairs for each domain: (i) Object detection: We fine-tune hundreds of YOLOv8l (Jocher et al., 2023) models on individual categories from Open Images V7, obtaining

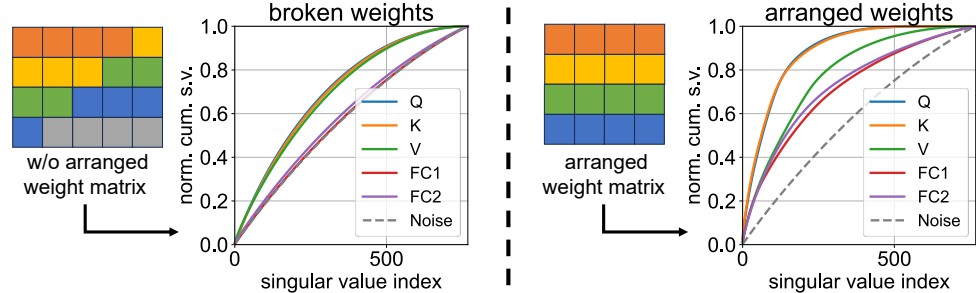

Figure 2: Comparison of normalized cumulative singular values for matrices of the first transformer layer in ViT-Base Dosovitskiy et al. (2021b). Each color in the illustration corresponds to a row in the weight matrix, and the gray squares denote zero padding. The steeper the curve (*i.e.,* the closer to the top-left corner), the stronger the low-rank properties of the matrix and easier to fit.

image-parameter pairs. (ii) Image generation: We train 1,000 LoRA adapters for Stable Diffusion 2 (Rombach et al., 2022), each on one ImageNet class, yielding style-guided image-parameter pairs. (iii) Image classification: Following RPG (Wang et al., 2025), we curate 1,022 classification tasks on CIFAR-10, forming embedding-parameter pairs.

## 2.3 STRUCTURE-AWARE ARRANGEMENT

**Inspirations.** The structure of neural network parameters is significantly more complex than that in images or texts. Both convolutional neural networks and transformer architectures first process local features or element-wise relationships in the input data (*i.e.*, images, texts, or others), then integrate this information into global representations through hierarchical structures. Here, a key question is: what is the local feature of neural network parameters?

**Key insight:** a layer of parameters can be conceptualized as a local feature in this context.

**Processing.** For a given layer (denoted as layer-$i$) within the network architecture, we analyze its corresponding parameters: the weight matrix $W_i \in \mathbb{R}^{d \times r}$ and the bias vector $b_i \in \mathbb{R}^r$. Beyond these raw parameters, we capture the statistical properties of each layer by calculating the mean and standard deviation of both weights and biases. Therefore, we can obtain normalized weights, biases, and statistical properties. This comprehensive processing can be formally expressed as follows:

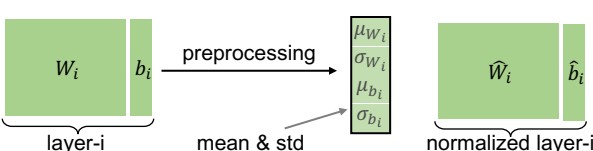

Figure 3: We obtain normalized weights, biases, and mean and standard deviation by preprocessing.

$$\{W_i, b_i\} \Rightarrow \{\hat{W}_i, \hat{b}_i\}, \; [\mu_{W_i}, \; \sigma_{W_i}, \; \mu_{b_i}, \; \sigma_{b_i}]^\top, \tag{1}$$

where $\hat{W}_i$, $\hat{b}_i$, $\mu_.$, and $\sigma_.$ denote the normalized weights, normalized biases, mean, and standard deviation, respectively. To better understand this process, we also show a simple pipeline in Figure 3.

**Stacking.** For each layer, we extract normalized weights and biases along with their corresponding means and standard deviations. We then stack these components across all layers to create comprehensive global representations of the entire model, as illustrated in Figure 4. To maintain consistent dimensions across these features, we apply appropriate padding operations to each layer. Through this structured approach,

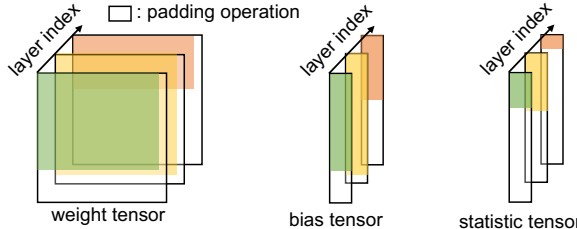

Figure 4: We stack weights, biases, and statistical measures as representation for model parameters.

we effectively represent the model parameters as stacked normalized weights, normalized biases, and statistical tensors. These arranged tensors serve as supervisions for the parameter generator.

## 2.4 SEMANTIC-PARAMETER DECODER

After arranging the supervisions, we now introduce the semantic-parameter decoder that transforms the data representation into model parameters. We note the data participated in the corresponding checkpoint collection as $D = [d_0, \cdots, d_j, \cdots, d_J]$.

**Data representation deformation.** We first use a pretrained model (*e.g.*, a vision transformer) to extract 2-D features tensors from input data $d_j$. As we stack all layers' parameters into 3-D tensors, we introduce a shape deformation module to obtain the 3-D representation of input data. This operation aims to obtain prototypes for the predicted parameters. As illustrated in Figure 5, the deformation module is simple, consisting of linear layers and reshape postprocessing.

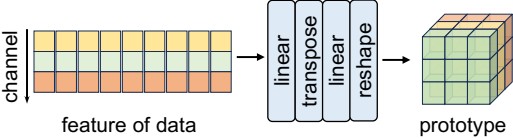

Figure 5: We apply a deformation module by linear layers and reshape to generate 3D parameter prototypes.

**Basic block of decoder.** Simple deformation modules alone cannot produce accurate predictions. Therefore, we design a decoder to comprehensively extract useful information from the prototype. Here, we present the basic block of our decoder. Each block extracts in-layer and cross-layer features in sequence. As shown in Figure 6, the in-layer transforming component uses two parallel branches with different orderings of convolutions across the height and width dimensions. After that, the cross-layer transforming component applies convolutions to combine information across the remaining dimensions. This basic block allows us to extract rich representations of data. Based on the basic block, we cascade $N$ basic blocks as the backbone of our decoder.

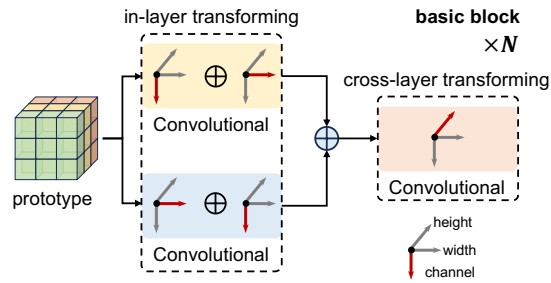

Figure 6: We design a decoder block that sequentially extracts in-layer features via two parallel branches of convolutions and cross-layer features by another convolution

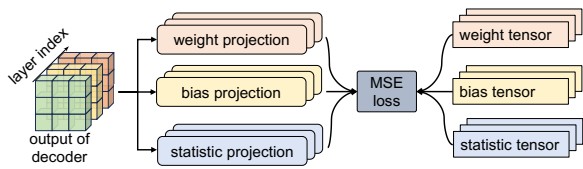

Figure 7: We put the output of decoder through specialized projectors into normalized weights, biases, and statistical measures. And the predictions are evaluated by MSE loss.

**Multi-head parameter projection** The decoder output contains rich feature representations. We utilize three specialized projectors to predict the normalized weights, biases, and statistical measures of the target model. As depicted in Figure 7, we implement separate projection paths for each parameter type and layer. The generated tensors are evaluated against ground truth values using MSE loss. Notably, these ground truths consist of normalized weights and biases along with their means and standard deviations. We subsequently denormalize predicted values to obtain final parameters.

## 2.5 TRAINING STRATEGY

Different layers have different sensitivities to the input condition changes. We found and analyzed this through sensitivity analysis of ViT-Tiny checkpoint dataset in RPG (Wang et al., 2025), as we observed significant variations in the std (sensitivity) of different layers to input condition changes (see Figure 8). To enhance the generator's responsiveness to these variations, we employ a weighted MSE-loss.

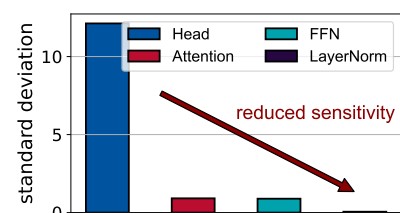

Figure 8: The standard deviations (std) of different layers across various tasks.

Specifically, we first compute the standard deviation of each parameter across all samples within each layer. These standard deviations are then normalized across the network. Next, we calculate an importance score for each layer by averaging the normalized standard deviations of all parameters within that layer. These layer-wise importance scores are used as weights in the MSE loss function, thereby emphasizing parameters that exhibit higher sensitivity to input condition variations (*i.e.*, those with higher standard deviations). Additionally, we introduce an adjustable exponent hyperparameter ($\alpha$) to control the intensity of the weighting effect. This allows for flexible tuning of how strongly the loss function emphasizes sensitive layers during training.

## 3 EXPERIMENT

### 3.1 SETUP

**Tasks, datasets, and architectures.** We evaluate our approach on diverse tasks using respective datasets: Open Images V7 (Kuznetsova et al., 2020) for object detection, ImageNet-1K (Deng et al., 2009b) for image generation, CIFAR-10 (Krizhevsky & Hinton, 2009) for image classification. The architectures involved in each task are reported in Table 1.

| task | dataset | architecture | # parameters |
|---|---|---|---|
| object detection | Open Images V7 (Kuznetsova et al., 2020) | YOLOv8l (Jocher et al., 2023) | 43.6M |
| image generation | ImageNet-1K (Deng et al., 2009a) | LoRA for Stable Diffusion 2 (Rombach et al., 2022) | 6.6M |
| image classification | CIFAR-10 (Krizhevsky & Hinton, 2009) | ViT-Tiny (Dosovitskiy et al., 2021b) | 5.6M |

Table 1: Overview of tasks, datasets, and model architectures used in our experimental evaluation.

**Training data collection.** For checkpoint collections, we fine-tune model on one single class of dataset for specified epochs, and save one checkpoint each time. This checkpoint is then combined with this class of data, forming class-specific data-parameter pairs. Through this process, we obtain abundant data-parameter pairs, which are then split for training and evaluating purposes.

**Training and inference details.** For the visual encoder that extract semantic features from images, we use ViT-H (Dosovitskiy et al., 2021b) in object detection and image generation tasks. For image classification, we directly feed binary embeddings into the decoder, hence no encoder is needed. The decoder then transforms semantic features into weight, bias, and statistic tensors, which are to calculate the weighted MSE loss with original parameters arranged by our structure-aware arrangement. During inference, encoder produces semantic features from images, decoder converts them to parameter volumes and tables. Then the output of the decoder are reconstructed into a complete neural network. Note that in inference, we require the generator to produce two types of parameters: seen parameters (*i.e.*, encountered in training) and unseen parameters (*i.e.*, not used in training). This would effectively measure both our method's reconstruction and zero-shot ability.

### 3.2 RESULTS OF PERSONALIZED PARAMETER GENERATION

#### 3.2.1 OBJECT DETECTION

**Results.**
We evaluated the mAP50-95 score distributions across seen and unseen parameter generation. We compared the collected, generated checkpoints, and generated models with additional 100 steps of fine-tuning's performance, as shown in Figure 9. It can be observed that: i) Most generated models achieve promising performance without additional training. This underscores our method's ability to generate high-performing parameters in both seen and unseen

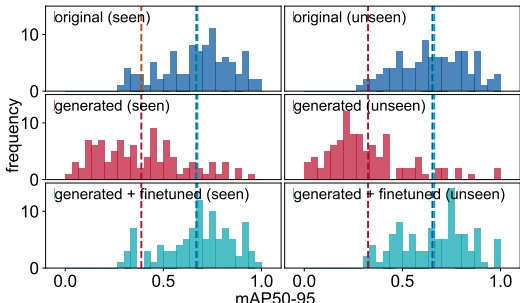

Figure 9: Our method surpass collected checkpoints with minimal fine-tuning both in seen and unseen cases.

scenarios. ii) Moreover, with merely 100 steps of fine-tuning, they surpass the performance of models fine-tuned over 1000 steps from pretrained checkpoints. It indicates our method can provide good initialization for model's training, especially for zero-shot datasets and tasks.

**Training vs. generating** Additionally, we evaluated the performance of traditional training and parameter generation in the setting of limited training data. Figure 10 shows generated parameters outperform traditional training under scenarios with very few training samples (*i.e.*, 1-8 images), while traditional training excels with more data (*i.e.*, 32+). This demonstrates parameter generation's advantage in scenarios with few shot training data, where traditional training can hardly converge.

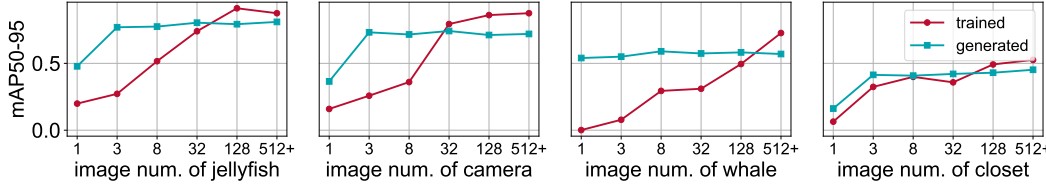

Figure 10: The sample-performance curve shows that our method outperforms traditional training when training samples are few, demonstrating that it is a powerful few-shot learner.

### 3.2.2 IMAGE GENERATION

Since there are limited real images (*i.e.*, ∼1,000 per class), we adopt Kernel Inception Distance (KID) (Bińkowski et al., 2018) to measure generated image's quality. We compare the performance of base SD2 (w/o LoRA), original LoRA adapters (original), and generated adapters. Results in Table 2 yields several findings: i) Our generated LoRAs

| KID↓ \ approach | without LoRA | original | generated |
|---|---|---|---|
| seen average | 0.062 | 0.055 | 0.043 |
| unseen average | 0.066 | 0.050 | 0.049 |

Table 2: KID scores (lower is better) of SD2 and original/generated LoRA on seen and unseen parts.

achieve comparable results to original ones, showing the design can effectively learning weight space distribution and generate high-performing parameters. ii) We also produce comparable parameters in unseen scenarios, indicating that this architecture is able to generalize to unseen parameters generation in inference.

Figure 11 shows detailed KID comparisons across unseen tasks. We can observe that generated LoRAs outperform original ones in most classes, demonstrating our method's effectiveness.

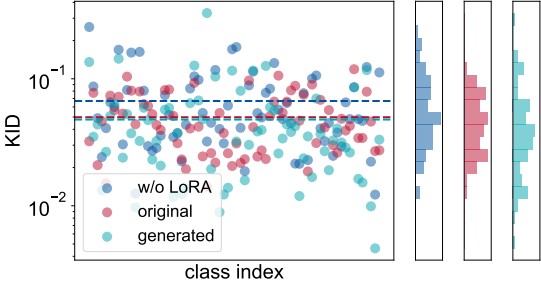

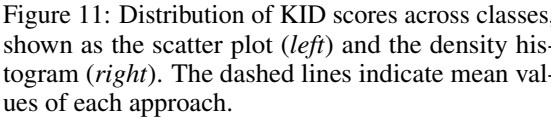

Figure 11: Distribution of KID scores across classes, shown as the scatter plot (*left*) and the density histogram (*right*). The dashed lines indicate mean values of each approach.

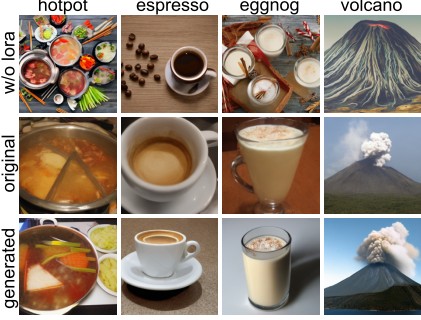

Figure 12: Visualizations of generated images. The above are the prompts for generation.

**Visualizations of generated images.** Figure 12 shows a qualitative comparison between three variants: w/o LoRA, original, and generated adapters. Our method synthesizes high-quality LoRA adapters without task-specific training, producing images more authentic compared to base SD2.

### 3.2.3 ON IMAGE CLASSIFICATION TASK

RPG (Wang et al., 2025) employs binary classification on the CIFAR-10 dataset to evaluate its generalization capability to unseen tasks. We follow the same setup to assess the zero-shot ability of our method. Specifically, we train multiple ViT-T that perform binary classification and pair them with respective embeddings to form the data-parameter pairs that are used to train our method.

**Results.** As shown in Table 3, our method consistently outperforms RPG across all unseen tasks while achieving results closer to the original model, demonstrating the effectiveness of our approach in generalization ability.

**Comprehensive comparison to RPG.** Beyond performance, we compare efficiency with RPG in Table 4. It shows that: i) Our method is substantially more efficient, with 7.1× smaller model size and 6.0× fewer training iterations compared to RPG. ii) During inference, our method achieves 183.8× faster speed and 2.5× lower memory cost, while maintaining a smaller performance gap (1.7% vs. 4.3%) to the original models.

| unseen tasks (embeddings) | | | | | | | | | | org. | RPG | ours |
|---|---|---|---|---|---|---|---|---|---|---|---|---|
| 0 | 1 | 0 | 0 | 0 | 1 | 0 | 1 | 1 | 1 | 97.3 | 94.4 | 96.4 |
| 0 | 1 | 1 | 1 | 1 | 1 | 0 | 1 | 1 | 0 | 98.1 | 96.6 | 98.8 |
| 0 | 0 | 1 | 1 | 1 | 0 | 1 | 1 | 1 | 0 | 97.4 | 95.0 | 96.8 |
| 0 | 1 | 0 | 1 | 1 | 1 | 1 | 1 | 1 | 1 | 98.4 | 96.1 | 97.7 |
| 0 | 0 | 1 | 0 | 0 | 0 | 0 | 0 | 0 | 0 | 98.9 | 96.6 | 98.8 |
| 0 | 0 | 0 | 1 | 1 | 0 | 0 | 1 | 0 | 1 | 96.7 | 92.9 | 96.3 |
| 1 | 1 | 1 | 1 | 1 | 0 | 1 | 0 | 0 | 1 | 97.6 | 94.8 | 96.9 |
| 1 | 0 | 1 | 0 | 0 | 0 | 0 | 0 | 1 | 1 | 98.1 | 95.7 | 98.2 |
| 0 | 1 | 0 | 0 | 0 | 1 | 0 | 1 | 1 | 0 | 97.1 | 93.6 | 96.3 |
| 1 | 1 | 0 | 0 | 0 | 1 | 1 | 0 | 0 | 1 | 97.0 | 94.0 | 95.6 |

Table 3: Our method consistently outperforms RPG, showing it is a more powerful zero-shot learner.

| efficiency item ↓ | RPG | ours | Impr. ↑ |
|---|---|---|---|
| training steps (K) | 120 | 15 | 8.0× |
| parameters (M) | 1263 | 431 | 2.9× |
| inference time (s) | 73.5 | 0.4 | 183.8× |
| inference memory (GB) | 6.2 | 3.2 | 1.9× |
| avg. gap from org. (%) | 4.3 | 1.0 | 4.2× |

Table 4: Our method surpasses RPG in terms of training cost, generation quality, and overall efficiency.

### 3.3 ABLATION AND COMPARISON

Unless otherwise noted, we carry out all ablation studies on image classification task. Note that to ensure evident comparison of different designs, we use a smaller decoder architecture (*i.e.*, 431M→122M parameters) compared with experiment in Section 3.2.3.

**The manner of structure-aware arrangement.** We compare our structure-aware arrangement with a sequential arrangement approach, *i.e.*, flattening and reshaping layer by layer, as well as the traditional training method. From Figure 13, our approach consis-

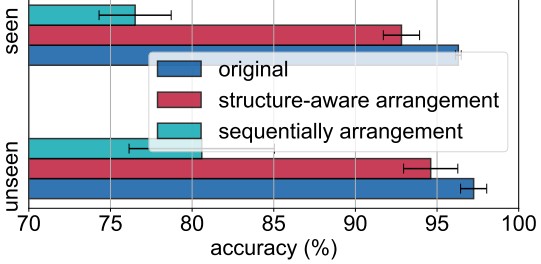

Figure 13: The structure-aware arrangement consistently outperforms the sequential arrangement and is comparable to the original.

tently outperforms sequential arrangement and achieves performance comparable to the traditional training method on both seen and unseen tasks. This indicates that preserving the important structural characteristics of parameters in the original network is of vital importance.

**Kernel sizes of hybrid blocks.** We now explore how convolution kernel size impacts model performance, as shown in Table 5a. Larger kernels generally result in improved performance, as small kernels have limited representational capacity and struggle to approximate the huge FC layer over the entire volume. However, too large kernels lead to increased computational overhead, as evidenced by longer training times. This analysis helps determine the optimal balance between model expressiveness and computational efficiency.

**Weighted MSE loss.** Table 5b shows how $\alpha$ in Section 2.5 affects performance. (i) With $\alpha = 0$ (no weighting policy applied), performance drops significantly compared to $\alpha > 0$, validating our weighted MSE loss. (ii) Performance peaks when $0.75 < \alpha < 1.00$. The results confirm weighted MSE loss helps the generator focus on important parameters, substantially improving performance.

**Number of training samples.** We also examine how the number of training samples affects generation quality as in Table 5c. When there is a limited amount of data, the model performs well

| k.s. | time (min) | accuracy (%) |
|---|---|---|
| 1 | 59 | $51.9 \pm 9.4$ |
| 3 | 69 | $90.7 \pm 3.6$ |
| 5 | 103 | $94.6 \pm 1.7$ |
| 7 | 155 | $94.8 \pm 1.9$ |
| 9 | 221 | $95.1 \pm 1.5$ |

| $\alpha$ | unseen | seen |
|---|---|---|
| 0.00 | $76.5 \pm 7.7$ | $77.8 \pm 7.4$ |
| 0.50 | $94.2 \pm 1.8$ | $94.2 \pm 2.3$ |
| 0.75 | $94.6 \pm 1.7$ | $94.9 \pm 1.9$ |
| 1.00 | $94.6 \pm 1.7$ | $94.8 \pm 1.7$ |
| 1.50 | $91.4 \pm 2.1$ | $92.5 \pm 2.1$ |

| #data | unseen | seen |
|---|---|---|
| 5 | $57.1 \pm 13.7$ | $97.5 \pm 0.9$ |
| 25 | $91.3 \pm 5.7$ | $95.8 \pm 1.8$ |
| 100 | $92.4 \pm 6.3$ | $94.5 \pm 2.2$ |
| 300 | $92.7 \pm 7.5$ | $95.5 \pm 1.7$ |
| 1002 | $94.6 \pm 1.7$ | $94.9 \pm 1.9$ |

(a) Larger kernel size (k.s.) can improve overall performance yet induces extra training costs.

(b) Weighted MSE loss improves our method's performance by balancing parameters' importance.

(c) Few samples are enough for seen parameter generation, but generalization requires more samples.

Table 5: We conduct experiments to validate three key aspects: (a) Kernel sizes of hybrid blocks: larger kernels improve performance but excessively large ones increase training costs; (b) Weighted MSE loss: it guides the generator to prioritize important parameters; (c) Number of training samples: more samples enhance generalization. Default settings are marked in gray. The tables show the average accuracies (%) across all tasks, with $\pm$ number indicating standard deviation.

on seen tasks but struggles on unseen tasks. It may be that the model overfits on the training data and lacks generality. As number of training samples grows, the model's generalization ability improves.

**Three-level hierarchy in basic block.** To determine which direction in our basic block is most critical, we conducted experiments by replacing the FC or convolution with linear interpolation along specific directions, blocking most information through those directions. We evaluated all combinations of blocked directions and report the results in Table 6.

The results indicate three points: (i) Every dimension in our basic block is crucial for performance, since neglecting them detriments performance. (ii) As basic operations, L dimension mixing can guarantee basic performance, as C and R dimension perform poorly when carried out alone. (iii) C and R dimension combined also yields good performance, showing this mixing strategy is effective.

| architecture | L, C, R | C, R | L, C | L, R | L | C | R |
|---|---|---|---|---|---|---|---|
| accuracy (%) | $94.6 \pm 1.7$ | $94.1 \pm 1.9$ | $93.0 \pm 1.9$ | $91.3 \pm 6.4$ | $78.2 \pm 11.3$ | $54.5 \pm 7.9$ | $54.6 \pm 10.3$ |

Table 6: Ablation study on three-level hierarchical mixing, confirming the distinct role of each dimension—layer (L), column (C), and row (R). The annotations are the same as in Figure 5.

### 3.4 COMPARISON WITH SOTA METHODS

We compare our method with the state-of-the-art methods and report the results in Table 7. It can be seen that our method outperforms all previous methods consistently. Moreover, even large-scale parameter generation methods like p-diff (Wang et al., 2024) and RPG (Wang et al., 2025) can suffer from out-of-memory (OOM) problems when the generated parameters are too large, but our method can still manage to generate comparable models. This indicates our method is efficient in generator size and memory usage.

| method | CIFAR-10 | | | | ImageNet-1K |
|---|---|---|---|---|---|
| | CNN (s) | CNN (m) | ResNet-18 | ViT-Base | ViT-Huge |
| params. (M) | 0.003 | 0.011 | 11.7 | 86.6 | 632.0 |
| $S_{\text{KDE30}}$ | 26.9 [46.1] | - | OOM | OOM | OOM |
| p-diff | 48.8 [49.0] | 61.9 [62.1] | OOM | OOM | OOM |
| SANE | - | 57.9 [57.2] | 68.6 [85.5] | - | - |
| D2NWG | 38.2 [44.7] | 58.8 [57.2] | 94.6 [94.6] | - | - |
| RPG | 49.0 [49.0] | 62.0 [62.1] | 95.1 [95.3] | 98.9 [98.7] | OOM |
| ours | 49.0 [49.0] | 62.0 [62.1] | 95.3 [95.3] | 98.9 [98.7] | 87.3 [87.8] |

Table 7: The blue subscripts denote the average accuracy of corresponding original models. And OOM denotes out-of-memory issue. Comparison with SOTA methods showcases our method has superiority in memory usage and generated model's performance.

## 4 RELATED WORKS

**Generative AI.** AI technologies have demonstrated remarkable performance across various domains, from recognition and object detection to segmentation (Lin et al., 2014; He et al., 2016; Vaswani, 2017; Dosovitskiy, 2020; Li et al., 2022). In recent years, generative AI (GenAI) attracts more and more attention from the research community, showcasing impressive capabilities in text, image, and video generation. The evolution of GenAI reveals a clear trajectory towards its ultimate goal: achieving personalized, controllable, and customized content generation. Early image generation methods, such as Goodfellow et al. (2014); Ho et al. (2020), were limited to synthesizing images from random noise, lacking the ability to control style, content, or other customizable features. Subsequently, various conditional image generation works (Dhariwal & Nichol, 2021; Ho & Salimans, 2022) have been proposed, enabling control over generated content and editing capabilities. Most recently, text-to-image methods (Rombach et al., 2022; Chen et al., 2023; Esser et al., 2024; Labs, 2024) have enabled image generation through flexible natural language descriptions. This evolution toward increased controllability and personalization is evident across various GenAI applications. This trend naturally raises an important question in the neural network parameter generation: is personalized generation capability equally crucial in this domain?

**Parameter generation.** The vision of parameter generation shares similarities with GenAI, with its core principle centered on learning from pre-trained checkpoints. Pioneering works include Stochastic neural networks (Sompolinsky et al., 1988; Bottou et al., 1991; Wong, 1991; Schmidt et al., 1992; Murata et al., 1994; Graves, 2011) and Bayesian neural networks (Neal, 2012; Kingma & Welling, 2013; Rezende et al., 2014; Kingma et al., 2015; Blundell et al., 2015; Gal & Ghahramani, 2016). These approaches initially focused on extracting prior knowledge from model weights to enhance model generalization and robustness. However, their application to large-scale generation tasks or complex real-world scenarios remains challenging. HyperNetworks (Ha et al., 2017) can synthesize various architectures' parameters through a small network. Smash (Brock et al., 2018) proposes a memory read-writes scheme to extend the range of synthesized architectures Recent years, several approaches (Peebles et al., 2022; Chou et al., 2023; Erkoç et al., 2023; Wang et al., 2024; Soro et al., 2024a; Lin et al., 2024; Li et al., 2024; Jin et al., 2024) use diffusion models to generate model weights. HyperRepresentations (Schürholt et al., 2022b;a; Schürholt et al., 2024) leverages autoencoder architectures to learn compact representations that capture the underlying distribution of model parameters. COND P-DIFF (Jin et al., 2024) and Tina (Li et al., 2024) advance the field by enabling parameter generation through task-specific conditions and language instructions, respectively. RPG (Wang et al., 2025) introduces a recurrent diffusion to generate hundred-million parameters and evaluate its generalization in unseen tasks. Despite these advances in parameter generation, considerable challenges still remain in achieving truly personalized parameter generation.

## 5 CONCLUSION

In this work, we present a novel approach that significantly advances real-time personalized parameter generation, marking a substantial progress toward practical usage of parameter generation in real-world cases. One of our core contributions lies in shifting the focus from learning parameter distributions to understanding the fundamental relationship between parameters and real-world data. Besides, our framework demonstrates exceptional personalized parameter generation capabilities, synthesizing tailored model weights based on given data and tasks within one second to seconds. We validate our novel paradigm by generating high-quality initializations followed by minimal fine-tuning – which reduces overall training costs to merely 10% compared to conventional approaches. In the future, we plan to enhance our arrangement techniques and decoder architecture to support network parameters with variable structures and symmetric properties. We are also interested in generating parameters that could be much better than the original ones.

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

# A    EXPERIMENT SETTINGS

We list the training recipes for reproducing.

| configure | object detection | image generation | image classification | ablation baseline |
|---|---|---|---|---|
| optimizer | AdamW | AdamW | AdamW | AdamW |
| learning rate | 3e-5 | 3e-5 | 5e-5 | 5e-5 |
| weight decay | 1e-4 | 0.2 | 2e-4 | 2e-4 |
| grad norm | 1.0 | 1.0 | 1.0 | 1.0 |
| training steps | 15000 | 15000 | 15000 | 7500 |
| batch size | 16 | 16 | 64 | 16 |
| learning rate schedule | cosine decay | cosine decay | cosine decay | cosine decay |
| condition noise aug. | 1e-3 | 1e-3 | 1e-3 | 1e-3 |
| target noise aug. | 2e-4 | 2e-4 | 2e-4 | 2e-4 |
| encoder | ViT-H | ViT-H | Linear | Linear |
| input | (1280,16,16) | (1280,16,16) | (256,4,4) | (256,4,4) |
| block-1 | (4096,32,32) | (384,128,2048) | (1536,32,32) | (256,32,32) |
| block-2 | (4096,32,32) | (384,128,2048) | (1536,32,32) | (256,32,32) |
| block-3 | (1280,128,128) | (384,64,1920) | (256,384,384) | (256,128,128) |
| block-4 | (434,640,640) | (256,32,1280) | (175,288,288) | (175,288,288) |
| output | (434,640,640) | (256,32,1280) | (175,288,288) | (175,288,288) |

Table 8: Training details of our experiments.

# B    ADDITIONAL EXPERIMENT RESULTS

## B.1    MORE RESULTS ON OBJECT DETECTION TASK

**Other evaluation metrics.** We make comparisons of mAP50, precision and recall between original (traditional training), generated (our generated model), generated+finetuned (finetune with our initialization) across different object categories in Figure 14.

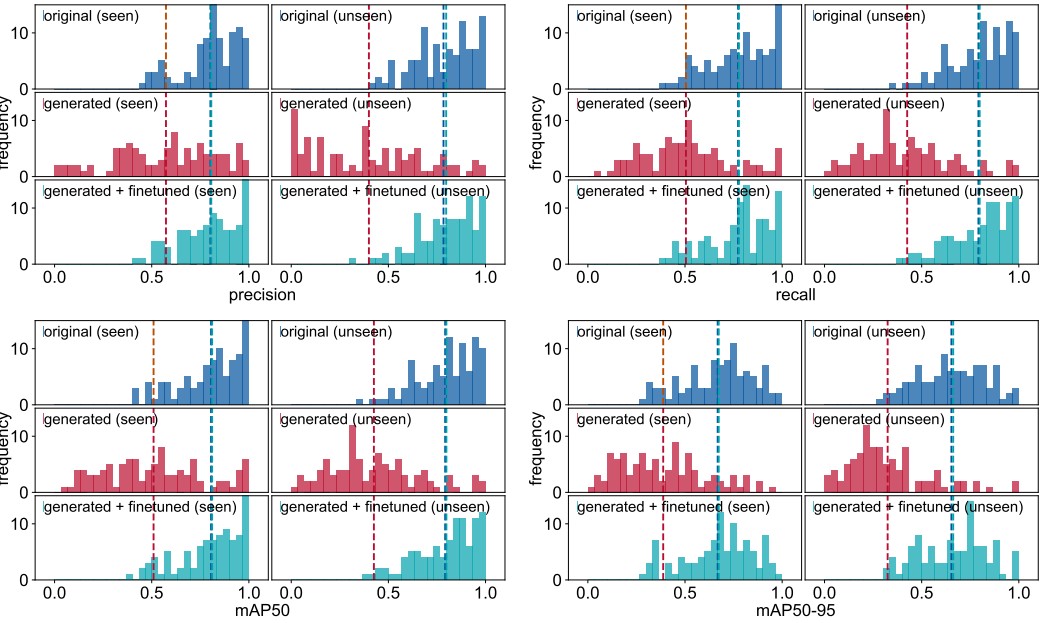

Figure 14: Our approach consistently achieves promising performance on both seen and unseen tasks, as evidenced by precision, recall, mAP50, and mAP50-95 scores. And our approach can always surpass the traditional training after fine-tuning 100 steps.

**Statistical overview.** We show the average precision, recall, mAP50, and mAP50-95 in Table 9, *i.e.*, the accurate values of the dashed lines in Figure 9 and Figure 14.

| | seen | | | | unseen | | | |
|---|---|---|---|---|---|---|---|---|
| | precision | recall | mAP50 | mAP50-95 | precision | recall | mAP50 | mAP50-95 |
| original | 0.805 | 0.776 | 0.808 | 0.671 | 0.787 | 0.770 | 0.797 | 0.655 |
| generated | 0.570 | 0.502 | 0.506 | 0.385 | 0.400 | 0.532 | 0.425 | 0.325 |
| generated + finetuned | 0.806 | 0.775 | 0.808 | 0.672 | 0.797 | 0.766 | 0.799 | 0.662 |

Table 9: Average precision, recall, mAP50, and mAP50-95 summarize performance across categories.

**Visualization.** In order to demonstrate the performance of our method, we visualized the results of object detection in Figure 15. Our method achieves promising results across various object categories.

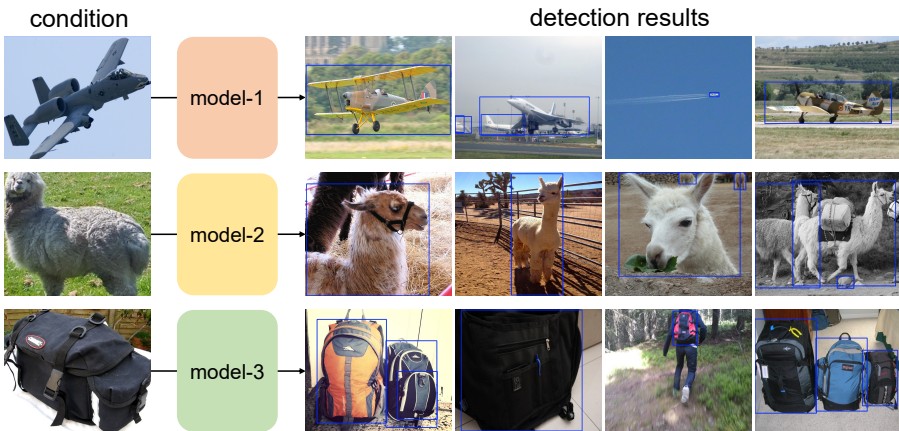

Figure 15: The conditions on the left are inputted into our method, and then our method generates the corresponding models. The results obtained using the generated models are shown on the right.

### B.2 MORE RESULTS ON IMAGE GENERATION TASK

**Evaluation with FID.** For image generation task, we also test Fréchet Inception Distance (FID) score, which is popularly used. But note that the number of original images in one category may less than 1000, probably leading to inaccurate FID scores.

| FID↓ \ approach | without LoRA | original | generated |
|---|---|---|---|
| seen average | 94.4 | 83.6 | 75.8 |
| unseen average | 102.2 | 81.9 | 85.7 |

Table 10: Comparison of FID scores for different methods across seen and unseen data.

**More visualization.** We present additional visualizations of the generated images in Figure 16. It can be observed that the images produced by our generated LoRA are closer to those generated by traditional training methods, meaning they better resemble the image style found in the ImageNet dataset. This indicates that our parameter generation model effectively follows the given conditioning information. It is important to note that our focus is not on comparing the diversity or overall quality of the generated images, but rather on evaluating the parameter generator's ability to adhere to the conditioning signals when generating LoRA parameters.

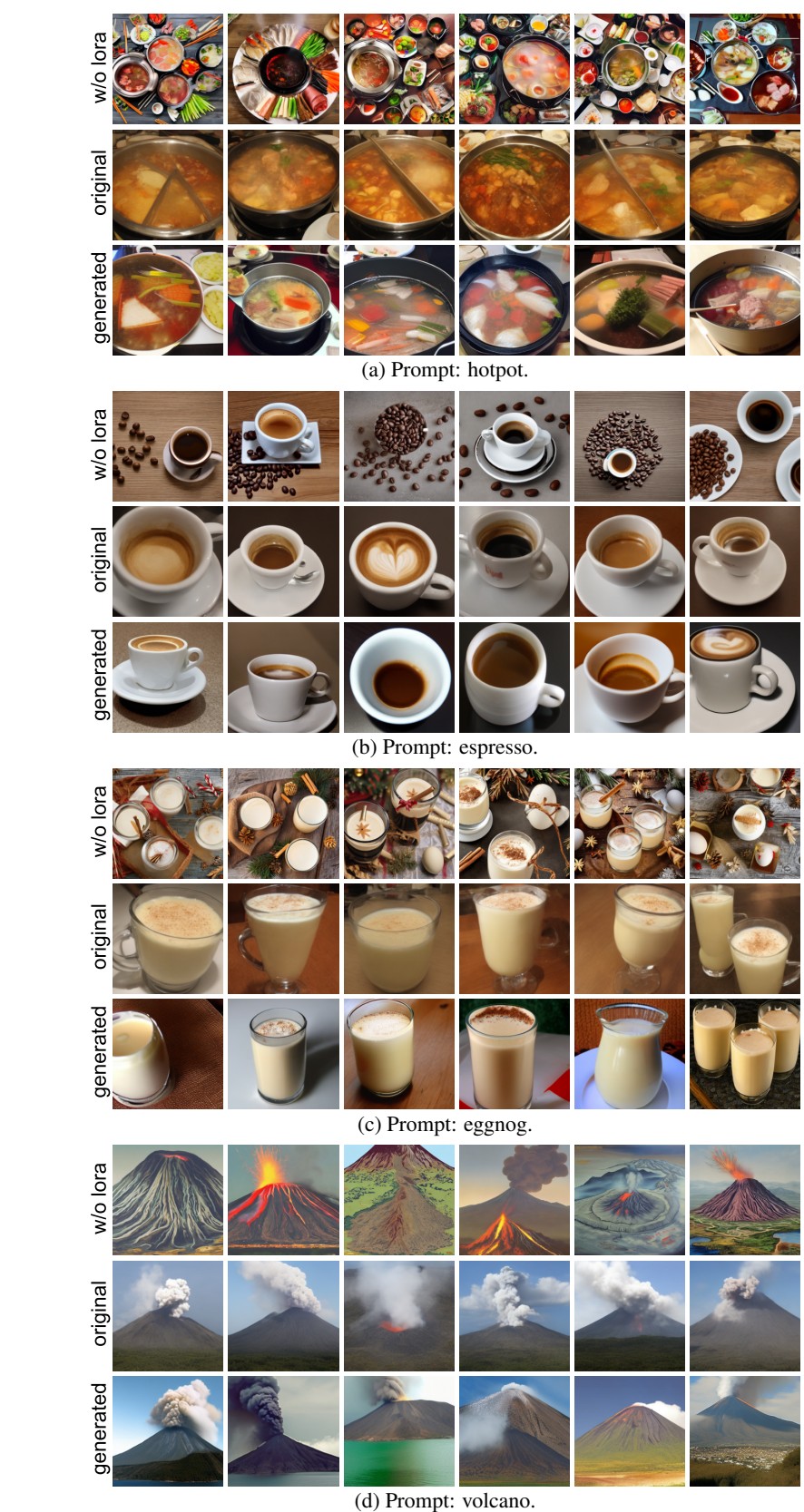

(a) Prompt: hotpot.

(b) Prompt: espresso.

(c) Prompt: eggnog.

(d) Prompt: volcano.

Figure 16: More visualizations of generated images.

## C    ADDITIONAL NOTES ON METHOD

### C.1    STRUCTURE-AWARE ARRANGEMENT OF VARIOUS MODULES

**Linear.** We model all types of neural network layers in a manner analogous to linear layers. The linear module itself is processed as illustrated in Figure 3, where normalized weights, biases, and relevant statistical properties are extracted. For other types of layers, we transform them into their equivalent linear counterparts, aiming to maintain a consistent modeling approach across the network.

**Convolution.** In convolutional neural networks, we take 2-D convolution as an example for analysis. For the 4-D weight tensor, (i.e., `[out, in, ks, ks]`. `out`: output channel, `in`: input channel, and `ks`: kernel size.), we have three flattening methods: `[out, in·ks·ks]`, `[out·ks·ks, in]`, and `[out·ks, in·ks]`. According to the analysis of Figure 2, we should choose the flattening method with the lowest rank, where fewer singular values capture the majority of the information. Therefore, we plotted the cumulative normalized singular values curve as shown in Figure 17. We observed that the three methods are nearly the same. For easily generalizing to 1-D and 3-D convolutions, we selected `[out, in·ks·ks]` as the default flattening method.

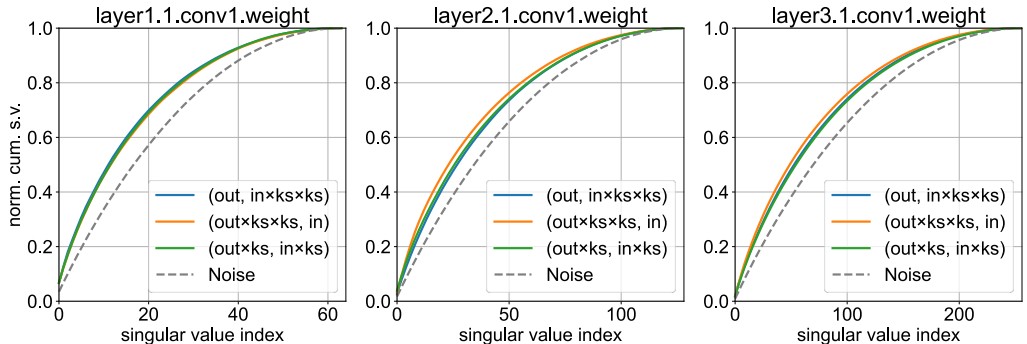

Figure 17: Caption

**Normalization.** For the normalization layer, it can be modeled as $\vec{y} = \vec{w} * \vec{x} + \vec{b}$, in which $*$ means element-wise multiplying. To transform it into its equivalent linear counterparts, we transform the modeling as $\vec{y} = \vec{x} \cdot diag(\vec{w}) + \vec{b}$, in which $diag(\vec{w})$ is the weight matrix in the linear counterpart.

### C.2    SEMANTIC-PARAMETER DECODER WITHOUT ACTIVATION FUNCTION.

In our parameter generation task, replacing traditional activation functions with LayerNorm provides a rational and effective approach by addressing distributional stability and enhancing nonlinear modeling. High-dimensional parameters are often information-sparse, which can lead to overfitting—especially when the amount of training data is limited.

LayerNorm helps mitigate this issue by preserving information across all neurons and facilitating smoother gradient propagation, enabling the network to learn underlying patterns more effectively even with limited data. Furthermore, LayerNorm's learnable parameters ($\gamma$ and $\beta$) allow for adaptive, nonlinear transformations, improving the model's expressive capacity. This adaptability may lead to superior performance compared to "ReLU" or "Tanh", particularly in capturing complex feature dependencies from small datasets.

### C.3    CONTRASTIVE FINE-TUNING OF THE IMAGE ENCODER

We train the encoder and decoder separately to stabilize training and enhance personalization while maximizing data utilization. The encoder leverages contrastive self-supervised learning Chen et al. (2020) on large-scale image datasets to develop robust and generalizable representations. The contrastive learning's criterion function is defined as follows, where $z_i$ and $z_j$ are feature representations

from the encoder, $\tau$ is the temperature (default 0.1), and $\mathbb{1}_{[k \neq i]}$ is an indicator function. The final loss is computed across all positive pairs, both $(i, j)$ and $(j, i)$, in a mini-batch.

$$\ell(i, j) = -\log \frac{\exp\left(\frac{z_i \cdot z_j}{||z_i|| \cdot ||z_j|| \cdot \tau}\right)}{\sum_{k=1}^{2N} \mathbb{1}_{[k \neq i]} \exp\left(\frac{z_i \cdot z_k}{||z_i|| \cdot ||z_k|| \cdot \tau}\right)} \tag{2}$$

$$\mathcal{L}_{\text{encoder}} = \frac{1}{2N} \sum_{k=1}^{N} [\ell(2k-1, 2k) + \ell(2k, 2k-1)] \tag{3}$$

### C.4 MSE LOSS DESIGNED FOR LoRA.

For LoRA, it consists of two matrices, $A$ and $B$. The product $A \cdot B$ is eventually added to the original network. However, for the a matrix $A \cdot B$, it is difficult to uniquely determine the individual matrices $A$ and $B$. Decomposing in different ways may lead to completely different matrices $A$ and $B$, which might also follow different patterns. This greatly increases the difficulty of learning parameter generation. Therefore, we consider computing the MSE loss based on $A \cdot B$, as shown in Equation 5, in which $L$ denotes the number of layers.

$$\text{For the matrix } X \in \mathbb{R}^{m \times n}, \text{ define } card(X) = m \times n. \tag{4}$$

$$\mathcal{L}_{\text{LoRA}} = \frac{1}{L} \sum_{i}^{L} \frac{\|A_i \cdot B_i - \hat{A}_i \cdot \hat{B}_i\|_2^2}{card(A_i \cdot B_i)} \tag{5}$$