# OpenReview forum: "Towards Personalized Parameter Generation via Data-Conditioned Mapping"
_ICLR.cc/2026/Conference — Submitted to ICLR 2026_

### Official Review · Reviewer_WqjL · 2025-10-17

**Soundness:** 2
**Presentation:** 1
**Contribution:** 1
**Rating:** 2
**Confidence:** 4

**Summary:**

The paper proposes generating LoRA adapter parameters conditioned on data, instead of fine-tuning. The goal is to enable fast, real-time model personalization by predicting adapter weights directly from input images.

**Strengths:**

Significance
1. The paper shows that parameter generation for LoRA adapters is computationally feasible at scale and can achieve real-time adaptation, suggesting potential for fast personalization systems.

**Weaknesses:**

Originality
1. The proposed approach is essentially a simplified adaptation of the RPG framework, which has already been accepted at NeurIPS 2025 [1]. Many of the core ideas, including the use of a normalization mechanism to stabilize parameter generation, are largely inherited from RPG. However, the authors do not provide a clear explanation of how their normalization differs from or improves upon RPG’s formulation. Without new theoretical insights, architectural innovations, or a detailed comparative analysis, the work appears incremental and mainly a reimplementation of prior work.

Quality and Clarity

2. The paper’s discussion of generalization is insufficient and potentially misleading. The “seen” and “unseen” settings appear to be defined only at the parameter level rather than at the semantic or data levels. This makes it unclear whether the method truly generalizes to new tasks or merely reuses existing representations from a pretrained backbone. The authors should clearly define how the “seen” and “unseen” settings are determined across datasets, and additional cross-dataset or out-of-distribution experiments are essential to validate the generalization claims. Moreover, the backbone and visual encoder are pretrained and fine-tuned on the same dataset used for evaluation. As a result, the validity of the claimed generalization is questionable.

3. The technical presentation of the paper is underdeveloped, poorly organized, and appears to have been written in a rush. The overall clarity is weak, and the work lacks sufficient methodological insight to support its proposed approach. (A) From a clarity standpoint, the explanations are shallow and imprecise, providing little insight into the underlying mechanisms or rationale behind design choices. (B) The normalization step, which appears to be a key component of the method, is mentioned but not defined or justified. It remains unclear whether it corresponds to L2 normalization, layer normalization, or another operation. Moreover, this normalization trick largely inherits the idea used in RPG, yet the authors neither reference RPG nor clarify how their formulation differs in terms of normalization, which undermines the originality and transparency of the approach. (C) In terms of technical depth, Section 2.5 lacks essential mathematical formulations and procedural details. The descriptions of the layer-wise importance scores, loss function, and hyperparameter tuning are vague, offering little technical insight into how the method actually works. (D) Regarding completeness, several statements are vague or confusing. For instance, the claim that “the structure of neural network parameters is more complex than that in images or texts” lacks clear meaning or conceptual grounding. In addition, important references such as SANE and D2NWG are missing, which further diminishes the paper’s technical clarity and overall quality.

Significance

4. The experiments are limited to image-based tasks and do not test whether the method generalizes to other modalities. Many reported results are demonstrations of the proposed approach without proper comparative baselines. Most comparisons are restricted to RPG. As a result, the empirical evidence for the paper’s claimed advantages remain weak.

5. The related work section is shallow and lacks meaningful engagement with prior research. It omits several relevant studies on LoRA parameter generation [2] and model merging [3], providing little context for how this work fits within existing parameter-efficient tuning methods. One paragraph is devoted almost entirely to a generic discussion of “generative AI,” which feels disconnected from the core technical topic. Without a deeper and more focused literature review, it is unclear how the paper advances the field or addresses existing gaps.

6. No source code is provided, which raises concerns about the reproducibility and transparency of the study.

[1] Recurrent Diffusion for Large-Scale Parameter Generation
[2] Conditional LoRA Parameter Generation
[3] Merging LoRAs like Playing LEGO: Pushing the Modularity of LoRA to Extremes Through Rank-Wise Clustering

**Questions:**

1. Please include an ablation study for the normalization trick and clarify what is being normalized and how important it is to performance. The authors must also include proper references to RPG, where this normalization idea was originally introduced.

2. Clarify the definition of “unseen,” specifying whether it refers to new image classes, different domains, or only unseen LoRA weights.

3. Provide explicit mathematical formulations and procedural details for the normalization, layer importance, training objectives, and other essential components. The authors should also explain the underlying insights and include proper ablation studies.

4. LoRA adapters could be trained with multiple images rather than with a single one. The paper should clarify how the images are paired with the pretrained LoRA adapters when training the parameter generation networks. Or is only one image used to train each LoRA adapter?

5. Add baselines and proper related works covering other LoRA-based parameter generation and model-merging techniques to better position this study within the current literature.

6. Provide clearer experiments to isolate the effects of pretraining and fine-tuning on the visual encoder to better analyze how these factors affect generalization performance.

---

### Official Review · Reviewer_EX73 · 2025-10-28

**Soundness:** 2
**Presentation:** 3
**Contribution:** 2
**Rating:** 4
**Confidence:** 4

**Summary:**

The authors propose a data-conditioned parameter generation approach to replace gradient-based fine-tuning for personalisation. The method features a structure-aware arrangement that preserves per-layer weight geometry, and a semantic-parameter decoder that maps data features to weights, trained with a sensitivity-weighted loss. Experiments on object detection (YOLOv8l/OpenImages), image generation (SD2 LoRA/ImageNet), and image classification (ViT-Tiny/CIFAR-10) claim <1-second personalisation and improvements over LoRA/RPG, with additional gains after fine-tuning.

**Strengths:**

The framework reports impressive inference efficiency (hundreds of times faster than RPG) and lower memory use. Several pragmatic design improvements are proposed, including the preservation of layer structure and sensitivity-based weighting. These changes are empirically supported through ablation studies.

**Weaknesses:**

W1. While the abstract and introduction emphasise sub-second adaptation, the experiments do not provide reproducible evidence for this. The only quantitative runtime reference (0.4 s inference vs. 73 s in RPG) is task-specific and lacks detail on hardware, batch size, or model scale. The heavier YOLOv8l and SD2 settings are not timed. The paper does not clearly describe how training data, checkpoints, and evaluation data are split. For example, do unseen categories share visual features with seen ones? Without this, the zero-shot results may be inflated. The process of “collecting hundreds of YOLOv8 models” is also vague and leads to reproducibility concerns.

W2. The method is described as generating personalised weights, but the evaluation benchmarks are not user or domain personalisation tasks. The “seen/unseen” task splits seem to correspond to classes rather than user or task-conditioned personalisation. The assumption is that paper, therefore, appears to measure data-conditioned interpolation rather than genuine personalisation, weakening its framing.

W3. Table 9 reports mAP50-95 = 0.385 (generated) vs. 0.671 (original) for seen data, and 0.325 vs. 0.655 for unseen. Only after an extra 100 fine-tuning steps do results reach parity with baselines. This implies that training-free parameter generation alone is insufficient for strong performance and contradicts the headline claim that adaptation is instantaneous.

W4. Although RPG is used as a primary baseline, other conditional weight generation models (e.g., COND P-DIFF, Text-to-Model, Tina) are cited but not extensively compared. Omitting them leaves unclear whether the improvements are specific to the structure-aware method or due to the task setup.

W5. The intuition behind structure-awareness and sensitivity-weighted loss is sound, but the paper lacks a formal analysis of why these improve generalisation or parameter distribution alignment outside of empirical observations in Figure 2.

**Questions:**

See weaknesses

---

### Official Review · Reviewer_ELFB · 2025-10-28

**Soundness:** 3
**Presentation:** 1
**Contribution:** 2
**Rating:** 4
**Confidence:** 4

**Summary:**

The authors tackle the problem of predicting parameters of a neural network that will work well on a specific dataset. The approach fits a neural network "generator" that is trained to predict these target-model specific parameters based on input datasets. They train this hyper-network on a selection of representative (dataset, checkpoint) pairs.
Their approach investigates how to best represent the regression target (NN parameters), a novel architecture of the hypernetwork itself as well as a targeted loss-formulation taking the sensitivity of target parameters to varying input conditions into account.
The authors show their method on vision-tasks, predicting up to 43.6M parameters for the YOLOv81 model.

**Strengths:**

The strength of the paper lies in its observation that the representation of the regression targets. As they show in 3.3, this change is highly impactful and intuitively relevant.

**Weaknesses:**

The weaknesses of this paper lie in the exposition. It is not clear exactly how their proposed hypernetwork is constructed and how it operates on an input dataset to produce a target network's parameters. Specific questions below.
Experimental evaluation: There are some open questions (below) about the experimental setup. Also I would like to see non-computer-vision experimental evaluations - otherwise the scope of this method is to narrow compared to what is claimed.
Related work: There is a field around HyperDreamBooth and follow-up work that the authors do not refer to. I would suggest replicating their setup for an additional "image generation" task in the experimental evaluation.

**Questions:**

Generator design:
How are the J 2D features transformed into J prototypes? Or just a single prototype? What is the underlying permutation invariant aggregation function?
Experimental design:
What is a "single class of dataset" for the training data collection? I'm assuming you're training three "generators", one per-task. What is the relation to "class" here?
Object Detection: What is the reference "pretrained checkpoint"? For "Training vs. generating", can you provide an aggregate metric across a larger selection of classes than just four?

---

### Official Review · Reviewer_63Jb · 2025-11-03

**Soundness:** 3
**Presentation:** 2
**Contribution:** 2
**Rating:** 4
**Confidence:** 3

**Summary:**

This paper introduces a data-conditioned parameter generation method capable of rapidly producing specialized model weights for given tasks. The core contributions include structure-aware arrangement, semantic-parameter decoder, and layer-wise learnable loss weighting scheme. The authors demonstrate the method's effectiveness across a range of applications, including object detection, image classification, and image generation.

**Strengths:**

1. The proposed method demonstrates significantly faster training and inference times compared to existing approaches, as shown in Table 4.

2. The method's effectiveness is comprehensively evaluated across a diverse set of tasks, including image classification, object detection, and image generation. The experiments encompass generalization test by including both seen and unseen data scenarios.


(minor)
The paper is well-illustrated with figures that effectively convey the intuition behind the proposed architecture. This aids the reader's understanding of the individual components of the method. However, this at the same time, leads to lack of explanation on some parts (see Q3).

**Weaknesses:**

1. A significant concern is the practical feasibility of the data collection process. The method requires a large dataset of "hundreds to thousands of specialized models", which appears to be an exceptionally demanding and potentially prohibitive prerequisite for real-world application. The paper would be strengthened by a more detailed discussion of the scalability and practicality of curating such a dataset.

2. The results in Table 5(c) suggest a potential limitation regarding generalization. While the method performs well on seen data distributions with few samples, its performance on unseen distributions appears to require a substantially larger number of samples. This raises questions about whether the model is overfitting to the training distribution of models/tasks. Further analysis on this trade-off would be beneficial.

3. The results presented in Table 9 and Figure 14 indicate that the generated parameters require additional fine-tuning to match the performance of conventionally trained models. This diminishes one of the method's key advantages by re-introducing a training step. The paper should clarify the practical benefits of this approach in scenarios where subsequent fine-tuning is necessary to achieve competitive performance.

**Questions:**

(Q1) Beyond faster inference, what are the primary advantages of the proposed method compared to standard fine-tuning? The significant cost of curating a large dataset of specialized models appears to trade the computational cost of model training for the effort of extensive data collection. A clearer articulation of the net benefit would be helpful.

(Q2) The proposed framework appears to be general. Could the authors comment on its applicability to other domains, such as NLP? What, if any, are the potential challenges or necessary adaptations for applying this method to other domains?

(Q3) (minor) The meaning of the values (0 and 1) in the leftmost column of Table 3 is unclear. The authors should clarify this in the main text or the table's caption.

---

### Meta-Review · Area_Chair_R4Lw · 2026-01-06

**Summary:**

The paper proposes a data-conditioned parameter generation method designed to replace standard gradient-based fine-tuning for real-time model personalization. The framework introduces a structure-aware arrangement, a semantic-parameter decoder, and a layer-wise learnable loss weighting scheme to generate model weights (such as LoRA adapters) directly from input data in less than one second.

Reviewers raised several critical concerns regarding the clarity of the methodology, the practicality of the required data collection (hundreds of specialized models), and whether the method truly generalizes to unseen tasks or merely performs interpolation.

**Reviewer Concerns:**

No rebuttals are provided.

**Reviewer Scores:**

No rebuttals and no discussions happened.

---

### Decision · Program_Chairs · 2026-01-26

Reject